# Enhancing Zero-Shot VLM Reward Models Through Structure-Aware Fine-Tuning

## Abstract

Designing effective reward functions remains a major bottleneck in Reinforcement Learning (RL). Recent work uses large foundation Vision-Language Models (VLMs) as zero-shot reward models, computing text–observation similarity to bypass manual reward engineering. Although promising, these rewards are noisy, brittle, and misaligned with ground-truth objectives. We introduce Structure-Aware Fine-Tuning (SAFT), a lightweight, LoRA-based method that adapts frozen VLM reward models online using simple structural priors. SAFT enforces invariances and proportionality in the reward signal via augmentations and auxiliary losses, yielding smoother and more consistent reward landscapes. Experiments across classic control and robotic manipulation tasks show faster policy convergence, substantially improved alignment with ground-truth rewards, and elimination of the extensive human annotation effort that Preference-based Reinforcement Learning (PbRL) would otherwise require. These results establish structure-aware fine-tuning as a simple path toward stable, text-conditioned reinforcement learning.

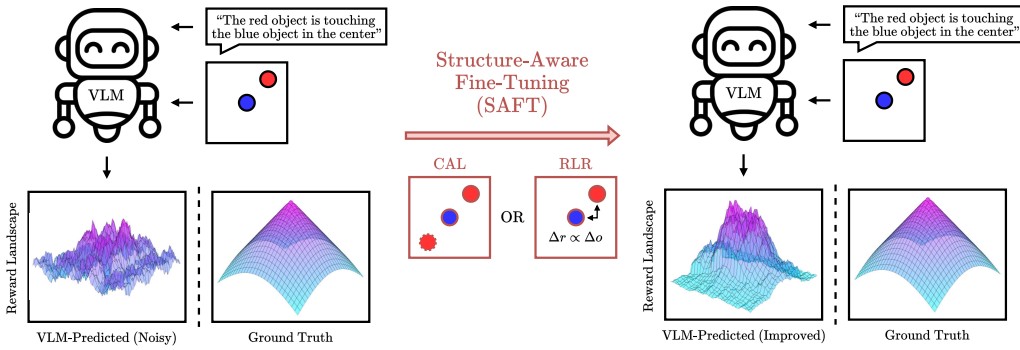

Figure 1: Structure-Aware Fine-Tuning (SAFT) improves the reward structure of zero-shot VLM reward models for reinforcement learning by enforcing invariances and proportionality, without any knowledge of the ground truth.

## 1 Introduction

Deep Reinforcement Learning (RL) has demonstrated remarkable success across a wide range of environments and applications (Vinyals et al., 2019; Levine et al., 2016). However, its reliance on reward functions presents significant practical challenges. As task complexity increases, designing effective state-based rewards becomes ever more difficult, often requiring substantial human effort and iterative refinement (Ng et al., 1999). Reinforcement Learning from Human Feedback (RLHF) (Christiano et al., 2023), and, more specifically, Preference-based Reinforcement Learning (PbRL) (Abdelkareem et al., 2022), offers an intuitive alternative by allowing humans to provide information about desired policies through comparative preferences that guide reward function learning. However, due to the sparse nature of preference feedback, considerable human oversight is still required (Hejna & Sadigh, 2022).

Recent work has explored leveraging advances in large Vision-Language Models (VLMs) to address these challenges by using these generalist models as a bridge between human intent and reward specification (Song et al., 2023; Xie et al., 2024). Large foundational VLMs employing shared embedding spaces for vision and language can be used as drop-in replacements for manually crafted reward functions by computing similarity between textual goal embeddings and visual observation embeddings. Previous work has applied these VLM reward models to derive dense rewards from sparse signals (Fu et al., 2024), improve learning efficiency from human demonstrations (Sontakke et al., 2023), and develop complex text-conditioned policies without requiring explicit rewards or demonstrations (Rocamonde et al., 2024). However, a critical limitation of these approaches is their dependence on VLMs to generate consistent goal-distance estimates without fine-grained task understanding, demanding generalized high-fidelity scene perception capabilities that current models cannot achieve (Sontakke et al., 2023; Rocamonde et al., 2024; Fu et al., 2024).

To address these limitations, we present *Structure-Aware Fine-Tuning (SAFT)*, demonstrating that VLM reward models can efficiently be fine-tuned in an online, self-supervised manner during reinforcement learning with minimal structural priors. Our approach enables solving previously intractable tasks, accelerates learning convergence, and improves alignment with ground-truth rewards when compared to non-finetuned baselines. Notably, SAFT delivers these benefits while dramatically reducing human labeling requirements relative to existing methods.

Our contributions are threefold. **(i)** First, we propose structure-aware fine-tuning using augmentations and auxiliary losses applied to VLM reward models during online training, achieving significant sample efficiency gains across a comprehensive set of environments. **(ii)** Second, we demonstrate substantial reductions in human labeling requirements through preference-based learning evaluation, while allowing smaller models to solve previously intractable tasks. **(iii)** Third, we show an improved alignment between fine-tuned VLM reward models and ground-truth reward functions, with benefits extending beyond online performance gains.

Our experiments demonstrate that SAFT enhances VLM reward models by improving sample efficiency, reducing human labeling requirements, enabling smaller models to achieve performance comparable to larger variants, and overall bringing them closer to the ground truth reward.

## 2 BACKGROUND

### 2.1 REINFORCEMENT LEARNING FROM HUMAN FEEDBACK

Reinforcement Learning from Human Feedback (RLHF) (Christiano et al., 2023) extends standard reinforcement learning to settings where the reward function is unknown, misaligned, or difficult to specify, inferring a reward model from human feedback instead of relying on a manually defined $R : \mathcal{S} \times \mathcal{A} \to \mathbb{R}$. The agent interacts with a partially observable Markov decision process $\mathcal{M} = (\mathcal{S}, \mathcal{A}, P, \Omega, O, \gamma)$, and in Preference-based Reinforcement Learning (PbRL) (Wirth et al., 2016; Christiano et al., 2023), learns from human preferences over trajectories $(\tau_1, \tau_2, \tau_1 \succ \tau_2)$ to train a parametric reward model $\hat{r}_\phi$. Using the Bradley-Terry likelihood

$$\mathbb{P}_\phi[\tau_1 \succ \tau_2] = \frac{\exp(\hat{R}_\phi(\tau_1))}{\exp(\hat{R}_\phi(\tau_1)) + \exp(\hat{R}_\phi(\tau_2))}, \quad \hat{R}_\phi(\tau) = \sum_{t=0}^{T} \hat{r}_\phi(s_t, a_t), \quad (1)$$

the negative log-likelihood is minimized over collected preferences

$$L_{\text{pref}}(\phi) = - \sum_{(\tau_1, \tau_2, \tau_1 \succ \tau_2)} \log \mathbb{P}_\phi[\tau_1 \succ \tau_2]. \quad (2)$$

The learned reward $\hat{r}_\phi$ is then used as a proxy for standard RL, enabling agents to learn aligned behavior from weak or indirect supervision in domains where direct reward engineering is impractical.

### 2.2 CLIP-TRAINED VISION-LANGUAGE MODELS

Vision-Language Models (VLMs) learn joint representations of visual and textual modalities. CLIP-trained VLMs (Radford et al., 2021) use dual encoders: a vision encoder $f_{\text{image}}$ and a text encoder $f_{\text{text}}$. Given an image $I$ and text $T$, the encoders produce embeddings $v = f_{\text{image}}(I)$ and $w =$

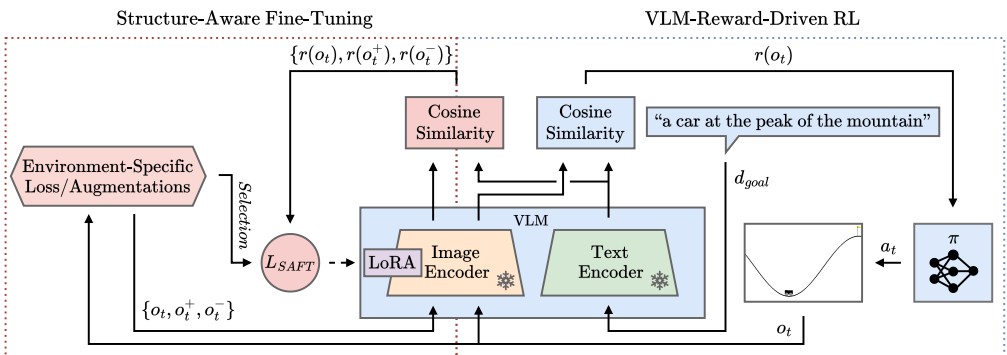

Figure 2: Structure-aware fine-tuning adapts VLM reward models online without any ground-truth rewards by updating LoRA modules with environment-specific augmentations and auxiliary losses, leveraging structural priors to improve alignment, sample efficiency, and reduce human feedback.

$f_{\text{text}}(T)$ in a shared $d$-dimensional space. Training maximizes cosine similarity for matching pairs and minimizes it for non-matching pairs via the symmetric cross-entropy loss:

$$L_{\text{CLIP}} = -\log \frac{\exp(v \cdot w/\tau)}{\sum_{i=1}^{N} \exp(v \cdot w_i/\tau)} - \log \frac{\exp(v \cdot w/\tau)}{\sum_{j=1}^{N} \exp(v_j \cdot w/\tau)}, \qquad (3)$$

where $\tau$ is a temperature parameter and $N$ the batch size. Trained jointly on large-scale image-text pairs without task-specific supervision, the shared embedding space enables zero-shot transfer by comparing image embeddings with textual prompts.

## 2.3 ZERO-SHOT VLM REWARD MODELS

Consider a goal-based reinforcement learning task where the agent's objective is to reach a target state. Instead of utilizing a manually crafted reward function, a textual description $d_{\text{goal}}$ specifies the desired outcome. At each time step $t$, the agent receives an observation $o_t$, which is passed through the VLM to obtain an image embedding $\boldsymbol{v}_t = f_{\text{image}}(o_t)$. The text encoder produces a goal embedding $\boldsymbol{w} = f_{\text{text}}(d_{\text{goal}})$, and the reward is defined as the cosine similarity between image and goal embeddings:

$$r_t = \cos(\boldsymbol{v}_t, \boldsymbol{w}) = \frac{\boldsymbol{v}_t \cdot \boldsymbol{w}}{\|\boldsymbol{v}_t\|\|\boldsymbol{w}\|}. \qquad (4)$$

This reward can substitute for the true reward in any RL algorithm without changes to the training loop, enabling policies conditioned on textual goals. Although the reward depends on observations, policy training still uses the underlying state $s$, consistent with prior work (Sontakke et al., 2023; Rocamonde et al., 2024; Fu et al., 2024).

## 3 STRUCTURE-AWARE FINE-TUNING FOR ZERO-SHOT VLM REWARD MODELS

### 3.1 STRUCTURE-AWARE FINE-TUNING

Previous work has shown that zero-shot reward models derived from foundational VLMs often fail catastrophically, with weak scene understanding and reward signals that collapse under even minor visual changes (Rocamonde et al., 2024; Fu et al., 2024). These brittleness issues make fine-tuning essential for building a stable and accurate reward landscape. Injecting inductive bias is a natural way to push models toward consistent behavior and stronger generalization.

Accordingly, we introduce Structure-Aware Fine-Tuning (SAFT), a lightweight LoRA-based (Hu et al., 2021) procedure that strengthens VLM reward models using auxiliary self-supervised objectives that encode the invariances and relational patterns of the target environment. SAFT implements this idea through two complementary auxiliary objectives: one that enforces invariance to

Figure 3: Contrastive Augmentation Loss enforces reward invariance for positive augmentations while increasing separation from negative examples, preserving task semantics and avoiding representational collapse. Visualization pertains to the ReposeCube environment.

task-preserving transformations and another that promotes proportionality between state changes and reward changes. Together, these losses yield reward models that respect environment symmetries, are less noisy, and align more closely with the true reward. We freeze the base VLM and insert LoRA adapters into the image encoder, fine-tuning them online with one of the two objectives described below. Figure 2 shows this process.

### 3.1.1 Invariance via Contrastive Augmentation Loss (CAL)

Reinforcement learning environments often exhibit structural symmetries that should be preserved in the reward function. For instance, in CartPole, a pole tilted 15 degrees left or right from vertical corresponds to equivalent states that should yield similar rewards.

We exploit these symmetries through task-specific augmentations. Positive augmentations $\{o_{t,i}^+\}_{i=1}^p$ are a set of $p$ transformations that should produce similar or equal rewards (e.g., horizontal flips in CartPole). Contrastive examples $\{o_{t,j}^-\}_{j=1}^n$ are a set of $n$ semantically distinct states that should yield different rewards. These can include hard negatives (e.g., vertical flips in CartPole) or softer negatives such as other time steps within the trajectory.

Given observation $o_t$, our *Contrastive Augmentation Loss (CAL)* enforces reward invariance to positive transformations while maintaining discriminative capacity:

$$L_{\text{CAL}} = \beta \, \text{std}\big(r(o_t), r(o_{t,1}^+), \ldots, r(o_{t,p}^+)\big) - (1-\beta) \, \text{std}\big(r(o_t), r(o_{t,1}^-), \ldots, r(o_{t,n}^-)\big), \quad (5)$$

where $r(o) = \cos(f_{\text{image}}(o), w)$, and $\beta \in [0,1]$ is a tunable weighting term. The first term minimizes variance among positive augmentations, enforcing invariance to semantically equivalent transformations. The second term maximizes variance among contrastive examples, ensuring the model maintains the ability to distinguish states and prevents representational collapse where all states receive similar rewards. For all experiments, we utilize $\beta = 0.5$, providing equal weighting to both terms. Ablation studies regarding this value are detailed in Appendix B.3.

We find that most environments naturally contain states with identical reward values, for example when camera viewpoints change, when background noise is present, or when states are equally related to the goal along different dimensions, making our augmentation-based approach broadly applicable across diverse tasks.

### 3.1.2 Proportionality via Reward Lipschitz Regularization (RLR)

In goal-directed environments, rewards often change smoothly with observations within a local region, reflecting the intuition that states closer to the goal tend to have higher rewards. We encourage this structural prior through *Reward Lipschitz Regularization (RLR)*, using a soft loss-based approach rather than the hard constraints typical of traditional Lipschitz methods (Gouk et al., 2020). A visualization of the effects of this auxiliary loss is shown in Figure 4.

Given a window of $W$ consecutive observations $\{o_i\}_{i=1}^W$, we compute the scalar mean reward $\bar{r}$ and the element-wise mean observation $\bar{o}$. We then define normalized rewards as $\tilde{r}_i = r(o_i)/\bar{r}$ and normalized observations as $\tilde{o}_i = o_i/\bar{o}$, where the division is also applied element-wise. With this,

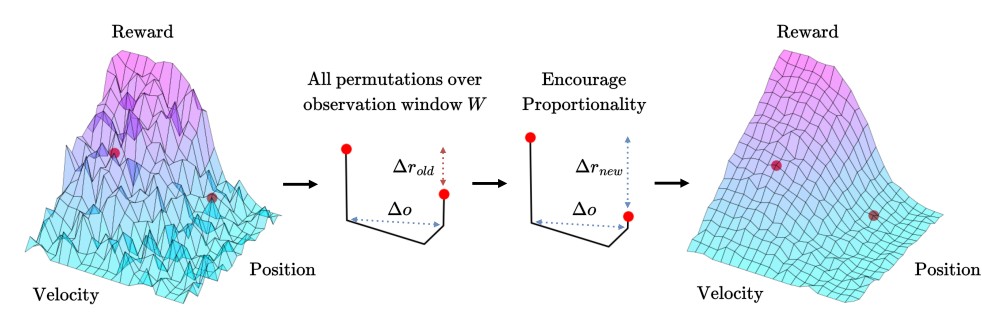

Figure 4: Reward Lipschitz Regularization (RLR) encourages proportional changes between observations and rewards, punishing large spikes and inconsistencies in the reward model. Visualization pertains to the MountainCar environment.

we define the Lipschitz regularization loss as:

$$L_{\text{RLR}} = \sum_{i=1}^{W} \sum_{j \neq i} \left( \frac{\|\tilde{r}_i - \tilde{r}_j\|_2}{\|\tilde{o}_i - \tilde{o}_j\|_2 + \epsilon} - 1 \right)^2 \tag{6}$$

where $\epsilon > 0$ avoids division by zero. This loss penalizes deviations from proportionality between observation distances and reward differences, encouraging the reward function to respect the underlying geometric structure of the state space. The window size $W$ determines the temporal scope over which proportionality is enforced. Larger windows impose stronger structural priors by requiring consistency across longer temporal horizons. Ablation studies regarding this value are detailed in Appendix B.4.

Although accurately measuring distances between observations in pixel space remains an open research problem, we adopt the L2 distance because of its simplicity and broad applicability within state-based RL. This regularization is effective across diverse environments, particularly where proximity in the observation space reflects progress toward goals. In higher dimensional settings where noise dominates the observations and the assumptions behind RLR are harder to satisfy, CAL is generally more suitable.

## 4 EXPERIMENTS

We conduct a variety of experiments to evaluate the efficacy of SAFT. Our evaluation examines which structural priors best match different environment types, whether online fine-tuning improves sample efficiency during policy learning, how much human labeling effort can be reduced when viewed through a preference-based learning lens, and to what extent our fine-tuned reward models align with ground-truth rewards.

### 4.1 MATCHING STRUCTURAL PRIORS TO ENVIRONMENTS

We evaluate our method across four goal-based RL environments: CartPole and MountainCar from the classic control suite (Brockman et al., 2016), where VLM reward models have shown prior success, and modified versions of Reach (Franka) and ReposeCube (Allegro) from Isaac Lab (Mittal et al., 2023), representing robotic manipulation tasks that have remained unsolved by VLM reward models without access to ground-truth knowledge. All learning is conducted solely with rewards provided by the VLM. The choice of auxiliary loss depends on the inherent structure of each environment. We select between CAL and RLR based on which structural properties can be exploited for better reward learning, as summarized in Table 1.

**CartPole**  CartPole exhibits both strong state-reward distance correlation and left-right symmetry (poles tilted $\pm\theta$ degrees left or right from vertical should receive equal reward). Therefore, we can apply both RLR to preserve distance relationships and CAL using horizontal flips as positives and vertical flips as hard negatives.

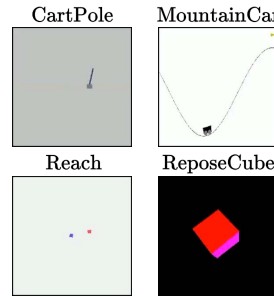

Figure 5: Visualizations of the four environments.

| | Classic Control | | Isaac Lab | |
|---|---|---|---|---|
| | CartPole | MountainCar | Reach | ReposeCube |
| CAL | ✓ | ✗ | ✓ | ✓ |
| RLR | ✓ | ✓ | ✗ | ✗ |

Table 1: Applicability of Contrastive Augmentation Loss (CAL) and Reward Lipschitz Regularization (RLR) across all four environments.

**MountainCar**  While maintaining a state-reward distance correlation suitable for RLR, the MountainCar environment lacks exploitable symmetries. Since the ground-truth reward is defined as distance from the right side of the screen, no two distinct states yield equal rewards, and hence no two observations can serve as positive augmentation pairs. We are therefore only able to apply RLR.

**Reach**  In the Isaac Reach task, the camera is mounted above the end effector, showing both current and target markers and centering the view so only relative goal motion is visible. The circular symmetry around the target (constant reward at fixed radius) violates RLR but provides rotational invariances exploitable by CAL. We use image rotations as positive augmentations and temporally distant observations as soft negatives.

**ReposeCube**  The Isaac ReposeCube task exhibits rotational symmetries similar to Isaac Reach. Multiple cube orientations yield equivalent angular distances to the goal, making RLR inappropriate. However, once again, these rotational symmetries provide natural invariances for CAL usage. As a result, we apply the same rotation-based contrastive augmentation strategy as in Isaac Reach.

Together, these environments provide a comprehensive baseline for VLM reward models, including sensitivity to asymmetries in dynamics (CartPole), unique states without valid augmentation pairs (MountainCar), rotational invariances that break distance-based rewards (Reach), and high-dimensional manipulation with non-trivial equivalences across cube orientations (ReposeCube).

### 4.2 POLICY PERFORMANCE

We first evaluate the sample efficiency of SAFT compared to baseline methods without fine-tuning, as well as goal-baseline regularization (Rocamonde et al., 2024) from prior work. We emphasize that our primary evaluation metric is the *relative* improvement yielded by SAFT over a base model, rather than absolute task performance. As SAFT is designed to refine existing reward signals, its utility naturally depends on the quality of the underlying VLM, a relationship we analyze across a broader spectrum in Section 4.4.

However, to capture the nuances of online training, such as convergence speed, stability, and sample efficiency, that scalar summary metrics cannot convey, we focus our main policy analysis on a representative regime where the base VLM is functional but imperfect. We initialize the VLM at a performance threshold where it can partially solve the task, allowing us to explicitly demonstrate how SAFT accelerates learning and stabilizes the reward landscape compared to the base model. This initialization is performed through pretraining using the ground-truth reward as detailed in Appendix A.3. This controlled setting isolates the benefits of our proposed method while also allowing us to use a computationally efficient 86 million parameter ViT-B-16 model (Ilharco et al., 2021) that achieves performance comparable to larger models without the associated computational overhead.

Our fine-tuning procedure uses LoRA adapters with 180K parameters, representing 0.2% of the total model parameters, consistent with standard practice. We evaluate performance across five seeds.

Figure 6 presents our policy performance results, showing that SAFT consistently outperforms the baseline methods across environments. These gains arise from correcting inconsistencies in VLM reward models, producing smoother and more accurate reward landscapes. The largest improvement in convergence speed is observed in CartPole, aligning with expectations as it provides a unique and

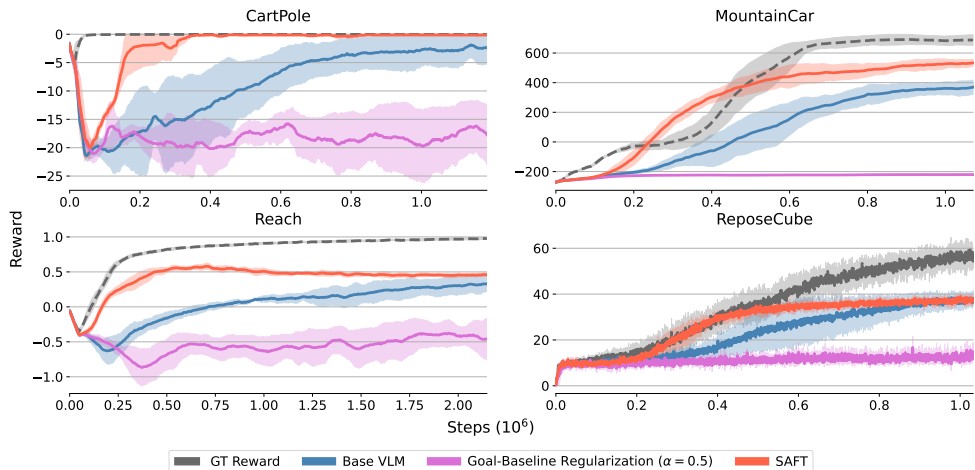

Figure 6: Policy learning curves comparing SAFT against the base VLM, goal-baseline regularization (Rocamonde et al., 2024), and Ground-Truth (GT) rewards across four environments. SAFT demonstrates superior sample efficiency and convergence speed, substantially reducing the performance gap to GT rewards. Shaded regions indicate standard deviation across five seeds.

| Environment | EPIC Distance ↓ | |
| --- | --- | --- |
| | Before SAFT | After SAFT |
| *Classic Control* | | |
| CartPole | 0.4306 | **0.2782** $\pm$ 0.0205 |
| MountainCar | 0.3106 | **0.2330** $\pm$ 0.0074 |
| *Isaac Lab* | | |
| Reach | 0.3443 | **0.3129** $\pm$ 0.0069 |
| ReposeCube | 0.2963 | **0.2605** $\pm$ 0.0098 |

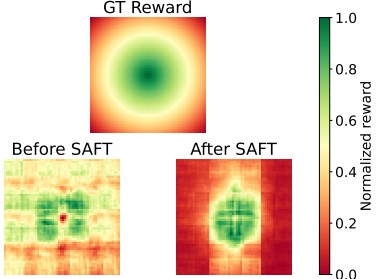

Table 2: EPIC distance between VLM reward models and the ground-truth reward. SAFT reduces EPIC distance, indicating closer alignment to the ground truth. Post-fine-tuning results report mean and one standard deviation across five seeds.

Figure 7: Reach reward landscapes predicted by the VLM with and without SAFT, showing improved alignment to the ground truth under SAFT.

unambiguous positive-negative example pair, unlike environments such as Reach and ReposeCube. Additionally, we find that in cases where the base VLM does not converge to the same value as the ground-truth reward, SAFT often reaches a higher final reward. This effect emerges from better shaping in the high-reward regime, enabling the policy to more clearly identify and exploit states corresponding to task success. In general, SAFT substantially narrows the performance gap between VLM-derived and ground-truth rewards.

## 4.3 REWARD MODEL ALIGNMENT

Next, we conduct offline evaluation to assess how well our fine-tuned VLMs align with ground-truth rewards compared to their unmodified counterparts. We employ the Equivalent-Policy Invariant Comparison (EPIC) (Gleave et al., 2021) distance as our evaluation metric, which measures differences between reward functions by comparing their canonical forms while removing the effects of reward shaping and scaling. This metric captures only the differences that affect optimal behavior, making it ideally suited for our analysis. For this analysis, we employ the base and fine-tuned VLM checkpoints from Section 4.2.

Table 2 presents the EPIC distance measurements on the different environments. Across all environments, fine-tuned models demonstrate substantially lower EPIC distances compared to their

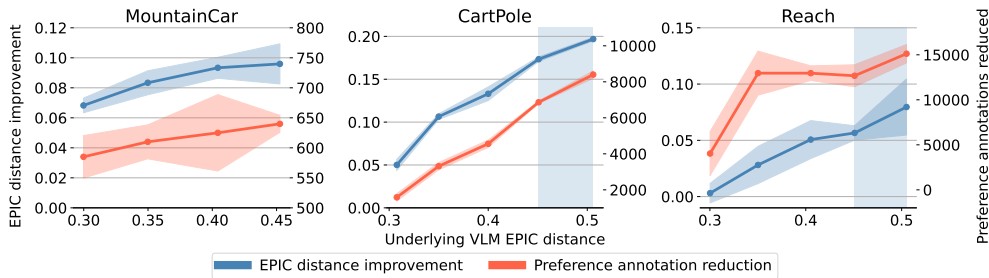

Figure 8: Utility of SAFT in relation to base VLM reward quality, averaged over two seeds. As the underlying VLM degrades, SAFT becomes increasingly useful. The critical regime, highlighted in blue, is where the base VLM fails yet SAFT enables successful training.

| | Classic Control | | Isaac Lab | |
| --- | --- | --- | --- | --- |
| | CartPole | MountainCar | Reach | ReposeCube |
| Binary Comparisons Saved ↑ | $5\,608 \pm 309$ | $670 \pm 27$ | $13\,516 \pm 1\,682$ | $8\,755 \pm 1\,711$ |

Table 3: Number of binary preference comparisons eliminated by using SAFT instead of human feedback, rounded to the closest integer. SAFT substantially reduces required human effort. Values show mean and one standard deviation across five seeds.

non-fine-tuned counterparts, indicating closer alignment with ground truth reward functions. These alignment improvements demonstrate that our method enhances fundamental reward quality across diverse task domains. Figure 7 shows VLM-predicted rewards before and after applying SAFT.

### 4.4 HUMAN LABELING EFFICIENCY

Finally, we quantify the utility of our approach through the lens of preference-based reinforcement learning (Christiano et al., 2023). In PbRL, human preferences constrain the hypothesis space toward behaviors aligned with those preferences, rather than letting the agent learn purely from unconstrained reward signals. Given the prevalence of this paradigm, we implement a PbRL baseline for direct comparison with SAFT.

For this baseline, we follow the same setup as SAFT by freezing the base VLM and fine-tuning the LoRA adapters within the image encoder. After each rollout, we generate binary preference queries and update the reward model online using the standard Bradley-Terry formulation as outlined in Section 2.1. As the reward model improves over time, we expect that after collecting a certain number of preferences, the resulting policy will reach the same performance as the SAFT-trained policy and hence the utility of the reward model will match that of SAFT. This enables us to quantify the amount of human labeling effort that SAFT avoids relative to this well-established alternative.

Table 3 reports the number of preference queries required by PbRL to match SAFT performance, showing savings of several hundred to several thousands of comparisons across environments. These results highlight two key advantages of SAFT. First, it reduces human labeling effort and avoids the cumulative effects of inconsistent human feedback, demonstrating both efficiency and robustness relative to human-in-the-loop approaches (Christiano et al., 2023). Second, SAFT introduces inductive bias through a one-time specification of augmentations, after which training proceeds automatically without further human intervention, simplifying the overall training process.

As stated in Section 4.2, the effectiveness of our method depends on the initial quality of the underlying VLM. An oracle VLM that perfectly models the ground-truth reward would render our improvements less impactful. To characterize this relationship, we evaluate binary feedback savings as a function of underlying VLM strength by applying SAFT to VLMs with varying levels of pretraining, as presented in Figure 8.

As the VLM reward model deviates from the ground truth, SAFT becomes increasingly valuable by correcting inconsistencies and reducing noise in the reward signal. We observe that SAFT consis-

tently improves performance across varying VLM strengths, lowering EPIC distance and effectively substituting for costly online human labeling. Notably, for most environments, there exists a critical regime in which the base VLM alone cannot solve the task due to noisy scene understanding, yet SAFT enables successful learning by mitigating this noise. Smaller variants within the same model family can fall into this regime and fail where larger counterparts succeed. Our method bridges this gap (Appendix B.1). Past a certain point, when the VLM is fundamentally misaligned with the task, our method cannot enable policy convergence, as the auxiliary losses operate without ground-truth supervision and are intended to improve consistency and reduce noise in the reward signal, but cannot compensate for a reward model that lacks any ability to interpret the scene.

## 5 RELATED WORK

**Self-supervised Auxiliary Losses for Reinforcement Learning.** Self-supervised auxiliary losses have long improved generalization and policy performance in RL (Jaderberg et al., 2016; Shelhamer et al., 2017; Yarats et al., 2020). Image augmentations are especially effective for vision-based policies (Laskin et al., 2020; Yarats et al., 2021), and contrastive learning methods such as CURL have also proven effective (Srinivas et al., 2020).

Another line of work has explored constraining networks to be Lipschitz continuous (Gouk et al., 2020; Scaman & Virmaux, 2019; Asadi et al., 2018). We take a similar approach but regularize the reward function rather than enforcing hard constraints and use L2 distances between states as a baseline despite the existence of more sophisticated metrics (Myers et al., 2025; Jiang & Qin, 2020).

So far, these auxiliary losses have been applied to policy networks. To the best of our knowledge, we are the first to adapt them for online fine-tuning of VLM reward models, combining Lipschitz regularization and visual augmentations to improve reward quality.

**Large Pretrained Models as Reinforcement Learning Reward Functions.** RLHF (Christiano et al., 2023) aligns policies with human intent by training reward models from preference data, but collecting such data is costly. Recent work instead uses foundation models for automated reward specification. Early methods had LLMs generate reward signals and refine them with policy feedback (Kwon et al., 2023; Song et al., 2023), while newer ones generate reward code (Xie et al., 2024; Ma et al., 2024; Li et al., 2025). However, the unimodality of LLMs limits these approaches to relying on the indirectness of intermediary textual representations.

Multimodal VLMs offer a promising alternative. Initial studies used them as success detectors (Cui et al., 2022; Du et al., 2023) or for generating preference labels in PbRL (Wang et al., 2024), though these approaches reduce information to binary signals or require auxiliary models. Others trained policies using embedding similarity (Mahmoudieh et al., 2022), but required large offline datasets.

Most closely related, Rocamonde et al. (2024) and Sontakke et al. (2023) use VLMs directly as reward models, with the former introducing goal-baseline regularization and the latter applying video-based models for motion goals. Fu et al. (2024) further show that fine-tuning with sparse ground-truth rewards improves dense reward quality. Our work builds on these directions, but improves policy training and reward model quality through simple structural priors, without requiring any knowledge of ground-truth rewards.

## 6 LIMITATIONS AND FUTURE WORK

Our main limitation is that the auxiliary loss, the augmentations for CAL, and the window size for RLR are chosen on an environment by environment basis, which reduces generality. However, this situation is not unique to our work. Early visual RL began with seemingly ad hoc augmentation techniques (Sadeghi & Levine, 2017; Lee et al., 2020; Cobbe et al., 2019), and over time, work such as RAD (Laskin et al., 2020) revealed randomized crops as a strategy that generalized well.

In the same spirit, we introduce ground-truth-free auxiliary losses for fine-tuning VLM reward models, playing a role for reward learning analogous to early augmentation studies for policy learning. SAFT shows that simple structural priors improve reward alignment, increase sample efficiency, and reduce human supervision. We do not claim a general purpose solution, but we view SAFT as a step toward more automated and broadly applicable approaches.

A natural next step is to replace manual choices with methods that automatically discover and combine structural priors within a unified framework that subsumes our current objectives. The text encoder remains frozen and underused, so future work could fine-tune it with contrastive goal descriptions and their textual negations to sharpen task understanding. As tasks grow in complexity, hierarchical scene decomposition and compositional augmentation may be required, for example in manipulation tasks that demand both rotation invariant grasping and position sensitive placement.

## 7    CONCLUSION

In this work, we introduce Structure-Aware Fine-Tuning (SAFT), a lightweight method for adapting frozen vision-language models whose purpose it is to provide zero-shot rewards for reinforcement learning. SAFT incorporates inductive bias through simple structural priors, applied via LoRA adapters to reshape and denoise the reward signal. This yields smoother and more consistent reward landscapes, faster policy convergence, closer alignment with the ground truth task reward, and the ability for smaller VLMs to solve tasks that otherwise require larger models. Our experiments across both classic control and robotic manipulation environments show that the brittleness and misalignment issues of general foundation models can be effectively mitigated by this approach, making VLM-based rewards substantially more reliable and practical. By addressing these core weaknesses, SAFT moves us one step closer towards the holy grail of fully text-conditioned policies.

ETHICS STATEMENT

This work does not involve human subjects or sensitive data. All experiments are conducted in simulation or on standardized robotic benchmarks, with no use of third-party or proprietary datasets. The research complies with the Code of Ethics of the venue. The proposed method provides a lightweight and general approach for fine-tuning vision-language reward models, improving alignment with ground-truth objectives and reducing reliance on costly human preference data. This can benefit reinforcement learning research by enabling more efficient, robust, and reproducible training without extensive human oversight. Potential risks include misuse of text-conditioned reward models to train agents for harmful goals, or unsafe deployment in high-stakes domains. These uses are not intended in this work, but the dual-use potential of text-conditioned reinforcement learning is acknowledged. To mitigate risks, all experiments are limited to simulation and benchmark tasks, and the method is designed only to improve reward stability and alignment with ground-truth objectives.

Large language models were employed exclusively for minor text polishing and were not used in any substantive capacity for research ideation or retrieval of related work.

REPRODUCIBILITY STATEMENT

The experiment results presented in this work can be reproduced with the implementation details provided in Appendix A. The code will be open-sourced on the project page upon acceptance.

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

| Component | Stable-Baselines3 | RSL-RL |
|---|---|---|
| Actor Network | MLP $[64, 64]$ | MLP $[256, 256, 256]$ |
| Critic Network | MLP $[64, 64]$ | MLP $[256, 256, 256]$ |
| Epochs | 10 | 1 |
| Batch Size | 64 | `rollout_steps` |
| Learning Rate | $3 * 10^{-4}$ | $10^{-3}$ |
| Gamma ($\gamma$) | 0.99 | 0.998 |
| GAE Lambda ($\lambda$) | 0.95 | 0.95 |
| Clip Range | $[-0.2, 0.2]$ | $[-0.2, 0.2]$ |
| Entropy Coefficient | 0 | 0 |
| Optimizer | `Adam` | `Adam` |

Table S4: Default reinforcement learning parameters from the Stable-Baselines3 (Raffin et al., 2021) and RSL-RL (Rudin et al., 2022) PPO implementations, used without modification in our experiments.

## A   IMPLEMENTATION DETAILS

### A.1   REINFORCEMENT LEARNING SETUP

For the MountainCar environment, we use the Stable-Baselines3 (Raffin et al., 2021) PPO implementation with default hyperparameters. For CartPole, Reach, and ReposeCube, we employ the RSL-RL (Rudin et al., 2022) PPO implementation, also with default hyperparameters. In all environments, the agent receives task reward exclusively from the VLM-based reward model without access to the ground-truth reward. Exact parameter values can be found in Table S4.

### A.2   VISION-LANGUAGE MODEL SELECTION AND LoRA FINETUNING

Following prior work, we adopt the ViT-B-16 backbone from the OpenCLIP (Ilharco et al., 2021) collection, pretrained on the LAION-2B dataset (Schuhmann et al., 2022), as our base vision-language model. The vision encoder is fine-tuned once per rollout using LoRA adapters (Hu et al., 2021), while the text encoder and all non-LoRA parameters of the vision backbone remain frozen throughout all experiments. The VLM reward serves as a drop-in replacement for the ground-truth reward, with the textual goal description kept fixed throughout training. Image observations are encoded by the vision backbone at every step and compared against this fixed goal embedding to produce rewards. VLM and LoRA configuration details are reported in Table S6.

### A.3   VISION-LANGUAGE MODEL PRETRAINING

To rigorously evaluate SAFT, we must isolate the method's structural contributions from the stochastic quality of the underlying Vision-Language Model. SAFT is designed to refine and structurally align *existing* reward signals. Therefore, its utility is naturally a function of the base model's capability. A base model that outputs pure noise cannot be meaningfully aligned, whereas a model that perfectly reflects the ground truth requires no refinement.

To evaluate SAFT on different points of this spectrum, as presented in Figure 8, we employ supervised pretraining as a controlled initialization protocol. This allows us to smoothly interpolate between these extremes, ranging from random initialization to near-oracle performance, and demonstrate that SAFT provides consistent relative improvements regardless of the base model's starting quality.

**Pretraining Methodology** To achieve these targeted initialization states, we pretrain the LoRA adapters by regressing the VLM reward predictions toward the ground-truth task rewards, as listed in Table S5. We employ Mean Squared Error (MSE) as the loss function:

$$L_{\text{pretrain}} = \|r_{\text{VLM}}(o_t) - r_{\text{scaled}}(s_t)\|^2 \tag{S7}$$

| Environment | Term | Weight |
|---|---|---|
| CartPole | $-\omega_{\text{pole}}^2$ | 1.0 |
| Reach | $-\|\boldsymbol{x} - \boldsymbol{x}_{\text{goal}}\|_2$ | 0.2 |
| | $1 - \tanh\left(\frac{\|\boldsymbol{x} - \boldsymbol{x}_{\text{goal}}\|_2}{\sigma}\right)$ | 0.1 |
| MountainCar | $x + 0.3$ | 1.0 |
| | $-a^2$ | 0.1 |
| ReposeCube | $(\Delta\theta + 0.1)^{-1}$ | 1.0 |

Table S5: Ground-truth reward definitions for the four environments. CartPole penalizes the pole's angular velocity $\omega_{\text{pole}}$. Reach penalizes the Euclidean distance between the end-effector $\boldsymbol{x}$ and the goal $\boldsymbol{x}_{\text{goal}}$. MountainCar rewards the car's horizontal position $x$, and punishes large actions $a$. Finally, ReposeCube rewards the agent based on the angular error $\Delta\theta$ between the cube's current and goal quaternions.

where $r_{\text{VLM}}(o_t)$ is the reward output of the VLM, given an observation $o_t$ at time $t$. Since this reward stems from calculating cosine similarity, and cosine similarity is strictly bounded to $[-1, 1]$, attempting to regress to arbitrary, unconstrained ground-truth values would prevent the VLM from learning the correct reward structure. To align the targets with the VLM's output space, we linearly scale the ground-truth reward $r_{\text{gt}}(s_t)$, given a state $s$ at time $t$, to the $[-1, 1]$ range:

$$r_{\text{scaled}}(s_t) = 2 \cdot \frac{r_{\text{gt}}(s_t) - R_{\min}}{R_{\max} - R_{\min}} - 1 \tag{S8}$$

where $R_{\min}$ and $R_{\max}$ represent the empirical minimum and maximum rewards of the environment. During pretraining, we apply this loss after every step.

We select regression over a PbRL-style pretraining objective for this phase because regression allows for linear, low-variance control over the reward structure (measured via EPIC distance). In contrast, PbRL pretraining tends to yield sparser and noisier rewards (Christiano et al., 2023; Tao et al., 2025), making it difficult to target specific capabilities for controlled evaluation. Furthermore, we posit that direct regression serves as the most faithful proxy for the behavior of future general-purpose VLMs, as it promotes a representation that remains unbiased toward any specific component of the reward structure. However, to demonstrate generality, we also provide an evaluation of SAFT applied to a PbRL-pretrained baseline in Appendix B.6.

**Selection of Operating Point for Main Results** While 8 demonstrates SAFT's utility across a range of model strengths, the detailed learning curves in the main text (Figure 6, Table 2, Table 3) utilize a specific initialization threshold where the base VLM is capable of partially solving the task. We identified this operating point empirically by rolling out the policy from different VLM pretraining checkpoints and visually checking when the policy showed learning and roughly solved the task. We deliberately selected this functional regime as relying solely on off-the-shelf models that often fail completely, as seen in the ViT-B-16 analysis in Figure S9, would reduce our evaluation to a binary success or failure outcome. By contrast, evaluating on a functional but imperfect model allows us to demonstrate that SAFT accelerates learning and denoises the reward landscape even when the base model is not catastrophic.

The exact pretraining budget for the base VLM in the main results of Sections 4.2 and 4.3 differs by environment. Cartpole uses 30 rollouts, resulting in $30 \times 32 = 960$ individual updates, or in other words 960 epochs of batch size 1. Reach uses $48 \times 27 = 1296$. MountainCar uses $100 \times 1$, because a full rollout would be excessive. ReposeCube uses $24 \times 180 = 4320$.

| Component | Configuration |
|---|---|
| *Vision-Language Model* | |
| Backbone | ViT-B-16 |
| Parameters | 86M |
| Source | OpenCLIP, trained on LAION-2B |
| *LoRA Adapters* | |
| Parameters | 180k |
| Target Modules | Feedforward Projection (`c_fc`), Output Projection (`c_proj`) |
| Rank | 2 |
| Scaling Factor | 32 |
| Dropout | 0.1 |
| Bias | None |
| Optimizer | `Adam` |
| Learning Rate | $10^{-6}$ |

Table S6: Vision-language model and LoRA training parameters used across all experiments.

**Strict Separation of Phases** We emphasize that this access to ground-truth rewards is strictly limited to the *pretraining phase* to establish experimental starting conditions. Once the SAFT fine-tuning phase begins, all access to ground truth is revoked, and the model relies exclusively on the self-supervised auxiliary losses ($L_{\text{CAL}}$ and $L_{\text{RLR}}$). Thus, no ground-truth information leaks into the SAFT update process.

**Off-the-Shelf Baselines** Finally, to complement the pretrained experiments, we provide training curves comparing SAFT against an off-the-shelf ViT-B-16 model without any pretraining for all four environments in Figure S9. These results confirm that even in regimes where the base model fails to converge, SAFT is useful, sometimes even being the deciding factor that recovers a solvable policy.

### A.4 ENVIRONMENT-SPECIFIC CONFIGURATIONS

We keep training parameters consistent across environments, introducing only minimal task-specific modifications. Experiments on CartPole, Reach, and ReposeCube are conducted within Isaac Lab (Mittal et al., 2023), while MountainCar is run in Gymnasium (Brockman et al., 2016).

For CartPole, all SAFT experiments are performed using CAL, reflecting the inherent left-right symmetry of the environment.

In MountainCar, we add an action penalty to stabilize learning and increase differentiation in the policy space so that the optimal policy is more clearly distinguished from alternatives.

The Reach task is modified to require only position alignment between the end effector and the goal, removing the orientation constraint. Rewards are computed from a top-down projection of proximity rather than full 3D distance, reflecting the limited depth perception of current VLMs. A visual tracker of the target and end effector positions is rendered to the agent.

For ReposeCube, instead of matching a held cube to a goal cube, the task is reformulated to require orienting the cube such that a specific side faces upward, with the cube itself rendered in view.

During CAL finetuning, CartPole employs a horizontal flip as the positive sample and a vertical flip as the hard negative. Reach and ReposeCube generate positives via rotation. Reach uses 8 samples at $\pm10°, \pm20°, \pm30°$, and $\pm40°$ along with a soft negative sampled from the same rollout. Repose-Cube uses 6 positive samples at $\pm10°, \pm20°$, and $\pm30°$ coupled with a hard negative rendered from the opposing $180°$ viewpoint.

For finetuning with RLR we use a window size of 32 for CartPole and 500 for MountainCar.

After each rollout, we compute one auxiliary loss, either $L_{RLR}$ or $L_{CAL}$, depending on the environment, and backpropagate it once. Gradients update only the LoRA adapters in the otherwise

| | Environments | |
|---|---|---|
| | CartPole | MountainCar |
| Rollout Steps | 32 | 1 000 |
| Parallel Environments | 64 | 9 |
| Auxiliary Loss | CAL | RLR |
| CAL Positive Sample | Horizontal Flip | N/A |
| CAL Negative Sample | Vertical Flip (*hard*) | N/A |
| RLR Window Size | 32 | 500 |
| Baseline Prompt | "pole and cart" | "a car in the mountain" |
| Goal Prompt | "pole vertically upright on top of the cart" | "a car at the peak of the mountain, next to the yellow flag" |
| | Reach | ReposeCube |
| Rollout Steps | 48 | 24 |
| Parallel Environments | 64 | 1 024 |
| Auxiliary Loss | CAL | CAL |
| CAL Positive Sample | Rotation ($\pm 10/20/30/40°$) | Rotation ($\pm 10/20/30°$) |
| CAL Negative Sample | Past Observation (*soft*) | Opposite Orientation (*hard*) |
| RLR Window Size | N/A | N/A |
| Baseline Prompt | "a red and a blue object" | "a cube with colors" |
| Goal Prompt | "the red and blue object are in the same place" | "only the pink side of the cube is visible" |

Table S7: Environment-specific parameters and augmentations used in all experiments. The labels *soft* and *hard* indicate whether the negative corresponds to an imperfect but plausible alternative observation or to a strict contradiction of the current state.

frozen vision encoder. To avoid leaking ground-truth information, we disable early stopping in all environments. Depending on task difficulty, we adjust the rollout length and the number of parallel environments. The exact parameters, CAL augmentations, as well as goal and baseline descriptions used in goal-baseline regularization (Rocamonde et al., 2024) are provided in Table S7.

## A.5 EPIC DISTANCE CALCULATION

EPIC distance, as defined by Gleave et al. (2021), is evaluated over a distribution of observations and their true rewards. To prevent skew within this distribution and improve the validity of the evaluation, we collect half of the observations using a random policy, and the other half from a policy trained using the true reward. In total, 2 000 observations are collected for evaluation. For ReposeCube, we directly render the cube, enabling us to synthetically generate 10 000 random rotations for EPIC distance evaluation.

## B ADDITIONAL RESULTS

### B.1 SAFT ENABLES OFF-THE-SHELF USE OF SMALLER MODELS

As discussed in Section 4.4, there exists a regime in which the underlying VLM alone cannot solve the task but succeeds when using SAFT.

Figure S9 illustrates the effects of SAFT when applied to an off-the-shelf ViT-B-16 model (Ilharco et al., 2021). We can observe that for the CartPole environment, the underlying model fails to converge unless SAFT is applied. This result highlights that enforcing the relative structure within the reward model via SAFT is not only beneficial for improving the efficiency of the sample in stronger models (Figure 6), but can also determine whether the training converges at all when using weaker ones.

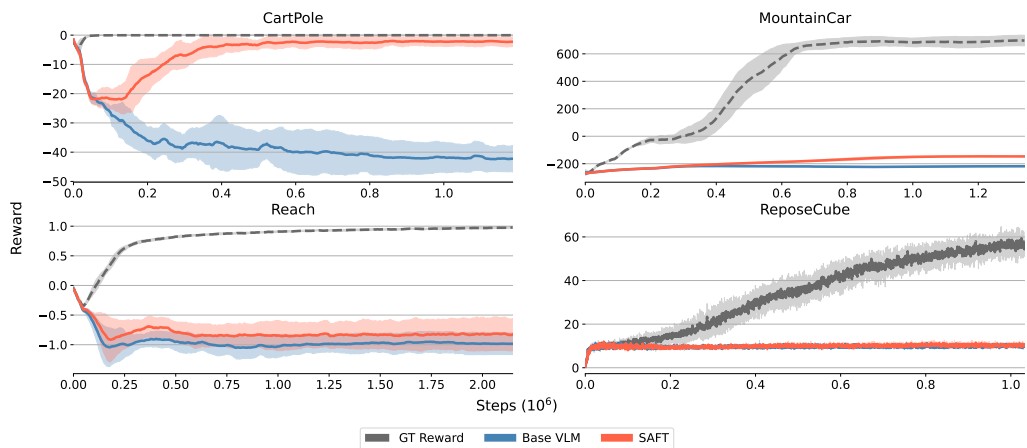

Figure S9: Results are averaged over five seeds for all environments, except ReposeCube which uses three. The off-the-shelf ViT-B-16 model fails to converge without SAFT, which shows that SAFT can be essential for convergence in smaller models.

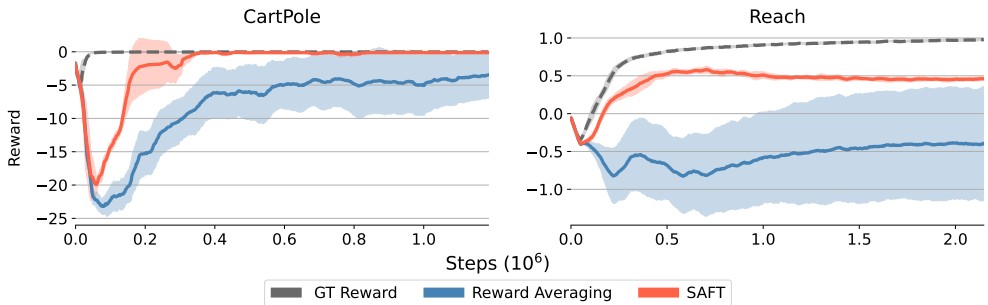

Figure S10: Comparison of SAFT with naive reward averaging over positive augmentations. Fine-tuning consistently outperforms averaging, indicating that simple smoothing falls short. Results presented over five seeds.

### B.2 NECESSITY OF FINE-TUNING

As fine-tuning the VLM incurs computational overhead, we first ask whether it offers benefits beyond a simple smoothing strategy that does not require any backpropagation. We compare SAFT, using CAL, to a naive baseline that averages the VLM's predicted rewards across positive augmentations without any fine-tuning. Figure S10 shows that fine-tuning consistently outperforms reward averaging, indicating that negative augmentations provide essential signal and that fine-tuning lets the model correct rewards for nearby states within the state space. This suggests that the gains come from learning rather than mere smoothing, and that the additional compute is justified.

### B.3 WEIGHTING OF CAL TERMS

The CAL objective has two components, a positive consistency term that minimizes variation in rewards across positive transformations, and a negative separation term that maximizes variation against negative examples. To assess the contribution of each term, we ablate the weighting parameter $\beta$ in Equation 5 by comparing our default $\beta = 0.5$ against $\beta \in \{1.0, 0.75, 0.25, 0.0\}$. The results presented in Figure S11 show that both terms contribute, with the negative separation term preventing representational collapse and the positive consistency term providing additional gains.

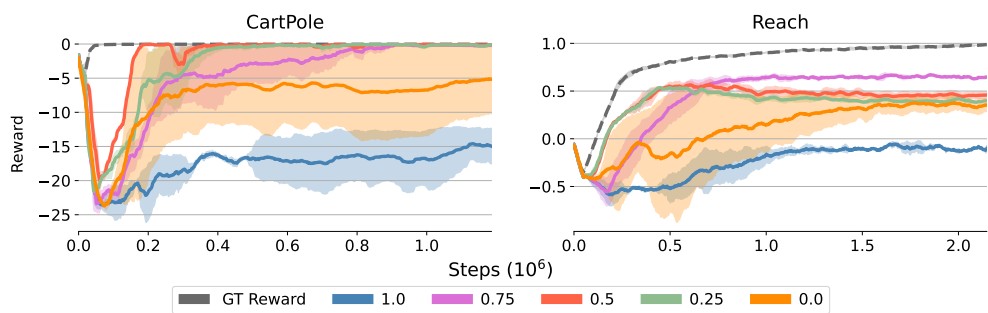

Figure S11: Ablation of the CAL weighting term $\beta$ between the positive-consistency and negative-separation terms, averaged over two seeds. Results indicate complementary contributions, with performance degrading when either component is downweighted.

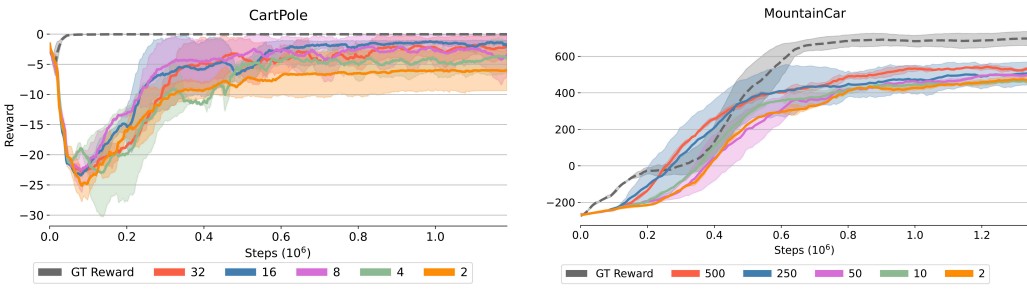

Figure S12: Ablation of the RLR window size $W$ determining the strength of the temporal consistency prior, averaged over two seeds. Results indicate that for both environments, larger window sizes tend to fair better.

### B.4 RLR WINDOW SIZE

The window size $W$ modulates the RLR temporal consistency prior, directly scaling the loss strength. We evaluate the sensitivity to this parameter by comparing our default settings against reduced window sizes: $W \in \{16, 8, 4, 2\}$ for CartPole (default $W = 32$) and $W \in \{250, 50, 10, 2\}$ for MountainCar (default $W = 500$).

Figure S12 demonstrates that larger window sizes improve performance in both environments. This suggests that the temporal consistency prior, which relates observation-space distances to reward-space distances, remains robust over long horizons.

### B.5 SIMULTANEOUS USAGE OF CAL AND RLR

The CartPole environment is unique in that it supports both RLR and CAL (Table 1). This lets us compare them directly and test whether their effects stack. Figure S13 shows that CAL outperforms RLR, likely because its positive and negative transformations are both precise and unique. Notably, although both methods are helpful on their own, their combination yields no further improvement.

### B.6 PRETRAINING USING PREFERENCE-BASED REINFORCEMENT LEARNING

As an alternative to the regression loss in Equation S7, we employ PbRL pretraining for cases where the initial VLM quality precludes solving the task. This approach requires no ground-truth rewards yet effectively primes the model for SAFT, as shown in Figure S14. We utilize the standard Bradley-Terry formulation of PbRL from Equations 1 and 2, performing updates after each rollout.

We note that the observed gains are smaller than those in Sections 4.2 and 4.3. This is consistent with Section 4.4, as the pretraining phase was longer and yielded a more capable base model.

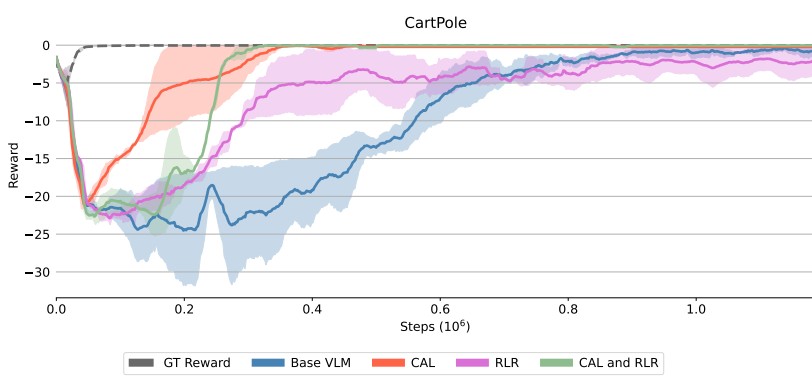

Figure S13: CartPole results comparing CAL, RLR, and their combination, averaged over two seeds. CAL outperforms RLR, and while each method is effective individually, their combination provides no additional gain.

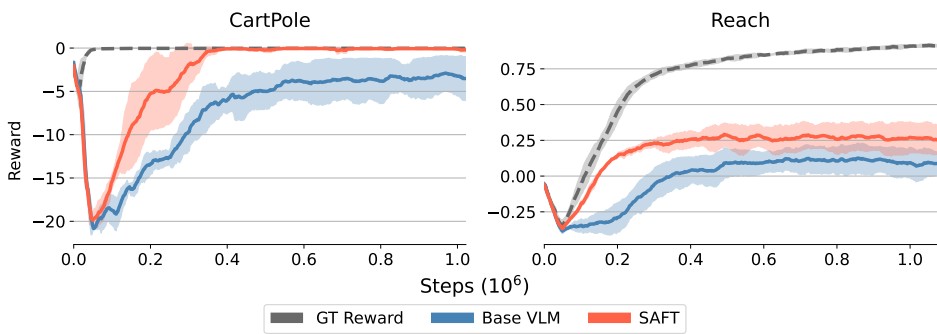

Figure S14: Application of SAFT to a model pretrained using a PbRL objective, averaged over five seeds. Results demonstrate that SAFT also yields significant performance gains following PbRL pretraining.

