# OpenReview forum: "Enhancing Zero-Shot VLM Reward Models Through Structure-Aware Fine-Tuning"
_ICLR.cc/2026/Conference — Submitted to ICLR 2026_

### Official Review · Reviewer_fyXP · 2025-10-25

**Soundness:** 1
**Presentation:** 1
**Contribution:** 2
**Rating:** 0
**Confidence:** 4

**Summary:**

The paper proposes Structure-Aware Fine-Tuning (SAFT) to improve zero-shot VLM reward models used in RL. A CLIP-style VLM provides dense rewards via image–text cosine similarity; SAFT inserts small LoRA adapters in the image encoder (text frozen) and fine-tunes them online during policy learning using simple structural priors. Two auxiliary, ground-truth-free objectives are introduced: Contrastive Augmentation Loss (CAL), enforcing reward invariance under task-preserving transforms, and Reward Lipschitz Regularization (RLR), encouraging proportional reward changes for nearby observations. SAFT is evaluated on CartPole, MountainCar, Reach, and ReposeCube; reported outcomes include faster policy convergence, reduced EPIC distance to ground truth, and large reductions in preference labels versus a PbRL baseline.

**Strengths:**

Clear high-level goal. Improve VLM reward reliability for RL without GT at policy time.

Very interesting idea. Structure-aware fine-tuning (SAFT) that encodes geometric/temporal invariants; plausible inductive bias.

**Weaknesses:**

As currently written, this paper lacks sufficient information to be reproduced or reviewed. The central issue is that the authors do not specify what loss function is used for pre-training the VLM for experiments in sections 4.2 and 4.3. The language of section A.2 (and in general, the rest of the paper) is highly ambiguous and extremely lacking in clearly-articulated details. Two sensible interpretations can be identified:

- Interpretation 1: The pre-training loss is the dense ground truth (GT) rewards the authors manually engineer for each task (which, incidentally, is only explicitly defined for some environments; but is not for the CartPole one, notably; also compromising reproducibility).
- Interpretation 2: The pre-training loss is their CAL/RLR objectives (which also require per-environment manual work, but are at least unsupervised).

Language by the authors would suggest the first interpretation ("This pretraining is performed online by regressing the VLM reward predictions toward the true task rewards."), which at least we know to apply to section 4.3; however, whether it also applies to results in 4.2 is unclear, as the grammar of this paragraph is poor.

If the method the authors used for their results is Interpretation 1 ("pre-training loss = dense GT rewards"), this would imply full leakage of the manually designed ground truth to the VLM, and invalidate all evidence the paper presents in support of their method and make most of their statements false, given the problem they purport to solve: avoiding the expensive and manual design of reward functions and replacing them with reward signals from vision-language models. At that point, in their experiments, all they show is that the VLM becomes, at best, a way to map from pixel observations to the state space, which is, granted, an interesting avenue of research, but not related to the claims of the paper, and requiring entirely different evaluation baselines. This would not, in principle, refute the validity of the method (it could still be a good method), but it would fully invalidate any evidence in support of it. Of course one would expect to find that such a method would "substantially improve alignment with ground-truth rewards", since the authors are training on the evals!

If, in contrast, Interpretation 2 is true ("pre-training loss = CAL/RLR objectives introduced by the authors"), then the baseline is just an ablation of the author's own method; claims should be framed as "online SAFT improves over offline SAFT-style pretraining" and the like; and, in any case, never "over zero-shot VLM", which is the central framing of the paper. There is nothing zero-shot about pre-training a VLM per-task, using per-task knowledge. Only if the pre-training were cross-task or, being per-task, used a loss function that depended in no way on knowledge about the task, would it make sense to talk about "zero-shot". Either way, the absolute non-negotiable baseline needed if Interpretation 2 is true would be to compare the results with a _frozen_ VLM (no pre-training). Assuming the results turned out to be positive (I find this plausible), I would still reject the vast majority of the claims and language used in the draft. A full re-framing of the results would be required to pass academic integrity standards, though could be still a very valuable contribution to the literature. Knowledge of geometric invariants about tasks is an extremely strong assumption when trying to jump from toy CartPole examples to general-purpose robotics tasks. The strength of VLMs for reinforcement learning is that knowledge about these invariants _emerges_ during pre-training due to the downstream generalization capabilities of foundation models. Any attempt to improve on this method should either (1) be comparably general or (2) clearly frame and explain the specific extent of applicability of the additional assumptions introduced. For example, for the latter, geometric invariants are cooked into physics-informed neural networks regularly, because the relationship between the assumptions made and the problem they solve is self-evident. To make the author's method valuable, they would have to either (1) significantly adapt their language, to make it clear to readers the obvious caveats of the approach instead of burying it at the end, or (2) provide a serious discussion of the extension capability of their method, which could well be, for all I know, widely extensible to more complex tasks, subject to evidence of it being provided.

The paper has a few other smaller issues, but those are inconsequential relative to the aspects outlined above.

If the authors were to entirely remove any per-task pretraining, updated all their experiments, and the results were still very strong (subject to seriously improving the writing with more clarity on reproducibility, added source code, etc.), my estimation of the paper's quality would significantly change; I'd rate it a 6 or 8, depending on specifics.

**Questions:**

Pretraining loss. Precisely what objective was used to “minimally pretrain” the VLM until it can solve tasks (GT regression? CAL/RLR? something else)? Provide the equation and targets.

4.2/4.3 pretraining budget. Total rollouts/steps, stopping criteria (“until it solves” is not reproducible), and any early-stopping or validation signals.

GT reward formulas. Give explicit mathematical definitions for each environment’s “true task reward” used in §4.3 pretraining (e.g., MountainCar “distance from right side,” Reach’s top-down projection). Include any shaping terms.

Frozen VLM baseline across all tasks. IIUC, you show off-the-shelf ViT-B-16 only for CartPole (2 seeds). Please add all 4 envs with ≥5 seeds.

CAL specifics used in main runs. exact counts/sampling for positives/negatives per step (p, n), any temporal distance thresholds for “soft negatives,” and the β value used in main results (not just ablation).

RLR normalization details. in Eq. (6), clarify how $\tilde{o}_i = {o_i}/{\bar{o}}$ is implemented for images (per-pixel mean over window? per-channel? scaling range?), and how pairs are batched when you “backpropagate once per rollout.”

---

> ### Author Response · Authors · 2025-11-21
> **Official Response to Reviewer fyXP (Part 1/2)**
>
> Its is clear you have spent a considerable amount of time trying to understand and review our paper for which we are very grateful! Below are our responses to the concerns, in the order they were raised.
>
> ### **1. Clarification of Scope and Pretraining**
>
> We begin by addressing the concern regarding ground-truth leakage. While your first interpretation of the pretraining setup is factually correct, the conclusion that this invalidates our results stems from a misunderstanding of our core contribution.
>
> You state that: "...this would imply full leakage of the manually designed ground truth to the VLM, and invalidate all evidence the paper presents in support of their method and make most of their statements false, given the problem they purport to solve: avoiding the expensive and manual design of reward functions and replacing them with reward signals from vision-language models".
>
> We would like to be very clear about this: we do **NOT** purport to solve the problem of avoiding expensive and manual reward design by replacing them with VLMs. This is something that the foundational work of VLM-RMs [1] have already done, and claiming this as our contribution would be incorrect and amount to plagiarism. We also do not believe that we ever claim this in our paper. SAFT aims to improve the reward landscape of existing VLM reward models without ground-truth rewards. The only metrics of interest for this are the **differences** between an existing VLM-RM and a SAFT-finetuned variant (or any non-ground-truth finetuning method, which to the best of our knowledge there are none, as we are the first to propose this). Our goal is not to show the solving of any task. Our goal is to show the difference in ability to solve the task before versus after using SAFT. We have adjusted our introductory claim on Line 69 to hopefully prevent future confusion.
>
> So why the pretraining? Our method of finetuning an existing VLM-RM is naturally dependent on the quality of the underlying VLM. Consider, for instance, a useless underlying VLM that only outputs a reward of zero at every point. No amount of SAFT finetuning will be able to meaningfully improve this VLM, as there is no way for the policy to understand what the goal is, and without any ground-truth reward knowledge, inferring this is impossible. On the other hand, consider a VLM that already perfectly models the ground-truth reward. Here too, SAFT will not be of use as the reward model is already perfect. Pretraining the VLM allows us to smoothly interpolate between these two extremes of underlying VLM capability and evaluate our method over the entire spectrum. In other words, pretraining is not a workaround, but a methodological necessity that allows us to give a fully comprehensive and unbiased evaluation of our method.
>
> **To reiterate, we want to make it abundantly clear that all comparisons and evaluations in the paper are fair and under no circumstances constitute "training on the evals", as this is not what we are measuring. We do not focus on absolute task performance. We care strictly about the improvement over the underlying VLM, regardless of its initial ability to solve the task.**
>
> With the previous points hopefully clarified, we would like to direct your attention to Figure 7, which demonstrates that SAFT consistently provides benefits across varying underlying VLM strengths. The specific results in Figures 6 and 8, along with Tables 2 and 3, isolate performance at a specific VLM capability level (Cross-referencing the EPIC distances in Table 3 with Figure 7 and the PbRL saved-comparison count confirms that this range aligns with Table 2).
>
> We selected this specific operating point because it represents the threshold where we empirically observed the underlying VLM could somewhat solve the task. We avoided presenting results solely using a weaker model (like the off-the-shelf ViT-B16 in Figure S9) to ensure we were not simply cherry-picking a "sweet spot" where the base VLM fails and SAFT succeeds. Instead, we demonstrate that even when the baseline VLM is capable of solving the task, SAFT remains highly effective, with improvements manifesting primarily in convergence speed. Dedicating a section to the focusing on a specific underlying VLM quality allows us to present detailed policy training curves, capturing the nuance of convergence speed and stability, rather than relying solely on scalar metrics as in Figure 7. We acknowledge this rationale was not explicitly stated in the original text and have amended the paper in certain sections (Lines 304-318, 851-860) to hopefully prevent future confusion. We thank you for this feedback!
>
> **References**
>
> [1] Rocamonde et al. "Vision-language models are zero-shot reward models for reinforcement learning" ICLR 2024
>
> [2] Tao et al. "Hybrid Reinforcement: When Reward Is Sparse, It's Better to Be Dense" arXiv preprint

---

> ### Author Response · Authors · 2025-11-21
> **Official Response to Reviewer fyXP (Part 2/2)**
>
> ### **2. Pretraining Details**
>
> Regarding our pretraining methodology and the modulation of VLM capability, we utilize regression to the ground-truth task rewards, whose definitions, as you correctly pointed out, are missing from the original submission. We apologize for this and have included it in the newest version, including shaping terms (Table S5). We choose to perform regression because it yields a smooth, linear improvement in EPIC distance, making the pretraining to specific EPIC distances (Figure 7) involve less guesswork. An alternative approach, and one we initially took, was to perform this pretraining using PbRL. However, the binary nature of preference-based feedback results in a much noisier and counterintuitive development of the reward structure, as it tends to sparsify rewards [2]. Furthermore, we posit that direct regression serves as the most faithful proxy for the behavior of future general-purpose VLMs, as it promotes a representation that remains unbiased toward any specific region of the reward landscape. Nonetheless, we have also included an example of SAFT demonstrating better performance over a PbRL-pretrained baseline in Appendix B.6 of the newest version of our paper. Lastly, we have dedicated a full section in Appendix A.3 to explaining the pretraining rationale and its details, including the budget.
>
> ### **3. Evaluation on Off-the-Shelf ViT-B16**
>
> As per your request, we have also provided training curves for SAFT versus the off-the-shelf ViT-B16 for all four environments, over five seeds.  We emphasize, however, that this off-the-shelf evaluation remains just "one piece of the puzzle", since our goal is not to show absolute performance, but rather only the difference SAFT makes compared to the underlying model.
>
> ### **4. CAL Specifics**
>
> We acknowledge that Table S7 was not sufficiently detailed and have expanded Appendix A.4 to clarify the selection and counts of positive and negative samples. We have also updated Equation 5 to explicitly show the β parameter and its default value of 0.5.
>
> ### **5. RLR Normalization**
>
> Thank you for pointing this out. Normalization is performed element-wise, and we have clarified this in Lines 215–216.
>
> ### **6. Reproducability and Code**
>
> Lastly, we fully align with your views, and take reproducibility very seriously as it is synonymous with academic integrity. As mentioned in our reproducibility statement, we will release the code upon acceptance, but we are also happy to provide it to you now for review purposes at: [Anonymous SAFT Source Code](https://anonymous.4open.science/r/saft-8447/). Please note however, that the current codebase is, as scientific code often is, messy and might be a little hard to navigate. We apologize for this lack of polish and assure you that we will strictly clean and document it for the final release. :)
>
> ### **7. Closing Remarks**
>
> As the main issue is hopefully cleared up, please let us know what the "few other smaller issues" are so we can address them as well.
>
> Once again, the large amount of effort spent reviewing and understanding our paper has not gone unnoticed, and we would like to thank you for this!
>
> **References**
>
> [1] Rocamonde et al. "Vision-language models are zero-shot reward models for reinforcement learning" ICLR 2024
>
> [2] Tao et al. "Hybrid Reinforcement: When Reward Is Sparse, It's Better to Be Dense" arXiv preprint

---

### Official Review · Reviewer_dAd2 · 2025-10-28

**Soundness:** 2
**Presentation:** 3
**Contribution:** 2
**Rating:** 4
**Confidence:** 4

**Summary:**

This paper introduces SAFT (Structure-Aware Fine-Tuning), a method for fine-tuning a Vision-Language Model (VLM) to produce more reliable reward signals (i.e., similarity scores) conditioned on visual observations and language-based task descriptions. The authors propose two self-supervised objectives, Contrastive Augmentation Loss (CAL) and Reward Lipschitz Regularization (RLR), to fine-tune the LoRA adapters of the VLM alongside policy training. The proposed method is evaluated on both classical control tasks (CartPole and MountainCar) and robotic manipulation tasks (Reach and ReposeCube), showing improvements over the original VLM.

**Strengths:**

- The proposed self-supervised objectives for fine-tuning VLMs seem novel and well-motivated, helping stabilize reward estimation in environments whose visual observations differ from the VLM's pretraining data.
- The descriptions and visual illustrations of the two objectives (CAL and RLR) are clear and make their roles in improving reward stability easy to understand.

**Weaknesses:**

- While the proposed objectives for fine-tuning VLMs are conceptually well-motivated by structural properties (symmetry and smoothness), the experiments are conducted only on simple environments that seem particularly suited to the proposed techniques. This raises concerns about the generalization ability and practical applicability of SAFT in more diverse or realistic settings. Prior works such as RoboCLIP [1] and VLM-RMs [2] demonstrated results in more complex environments (e.g., humanoid control or robot arm manipulation), yet it remains unclear whether SAFT can perform effectively in such settings or whether crafting augmentations in these environments would be as straightforward.

- Among closely related works (RoboCLIP, VLM-RMs, FuRL), only VLM-RMs is included for comparison. Additional comparisons, especially with FuRL, which also fine-tunes VLMs, are necessary to more comprehensively evaluate the effectiveness of the proposed method.

- The paper mentions that states differing only by viewpoint changes or background noise (L199–L202) should have identical reward values, however, no experiments involving corresponding augmentations or transformations are provided to support this claim.

- Lack of analysis for different window sizes $W$.

- In Section 4.2, it is unclear how the paper performs "minimal pretraining" of the VLM such that the base VLM model can solve the tasks (L295–L297). This point remains ambiguous even after reading the Appendix (L740–L742). Which loss function is used for fine-tuning at this step?

- The paper claims that the proposed method substantially improves alignment with ground-truth rewards; however, as shown in Figure 6, among the four environments, only CartPole shows a noticeable match to ground-truth reward performance.

Minor comments:
- The EPIC distance should be introduced in Section 4.3 rather than Section 4.4 to provide better context for L375–L377 and Figure 7.


References:

[1] Sontakke, Sumedh, et al. "Roboclip: One demonstration is enough to learn robot policies." NeurIPS 2023.

[2] Rocamonde, Juan, et al. "Vision-language models are zero-shot reward models for reinforcement learning." ICLR 2024.

[3] Fu, Yuwei, et al. "FuRL: Visual-language models as fuzzy rewards for reinforcement learning." ICML 2024.

**Questions:**

1. How does SAFT perform in more complex settings, as evaluated in RoboCLIP or VLM-RMs? How does SAFT compare to RoboCLIP and FuRL?

2. Could you clarify how the base VLM is pretrained such that it can solve tasks without SAFT?

3. Why does the Goal-Baseline Regularization baseline (VLM-RMs) perform worse on CartPole? In VLM-RMs's paper, this baseline showed perform well on CartPole, and also evidenced in [4].

4. Could you provide an analysis for different window sizes $W$?

5. Is there any factor that controls the strength of RLR during fine-tuning?


References:

[4] Wang, Yufei, et al. "RL-VLM-F: Reinforcement learning from vision language foundation model feedback." ICML 2024.

---

> ### Author Response · Authors · 2025-11-21
> **Official Response to Reviewer dAd2 (Part 1/2)**
>
> We thank the reviewer for their time and the extensive feedback! Below are our responses to the concerns, in the order they were raised.
>
> ### **1. Environment Complexity and Suitability**
>
> We respectfully disagree that our evaluation is limited to simple environments. Contrary to your claim, we do indeed evaluate on robot arm manipulation via the Reach task (IsaacLab Franka robot). Regarding the humanoid comparison: VLM-RM evaluates on a modified humanoid task requiring the agent to "fall" into a static pose (e.g., "kneel") because frame-based VLMs cannot encode motion targets (e.g., "walking"). This modification lacks a standard ground-truth reward function, forcing those authors to rely on qualitative empirical evaluation. This prevents reliable quantitative comparisons of policy performance or EPIC distance. Additionally, as qualitative evaluation is highly susceptible to human bias, we purposefully omit this task.
>
> To ensure rigorous evaluation on a high-dimensional task, we instead utilize ReposeCube. This task features observation and action spaces of comparable dimensionality to the Humanoid task ((72,16) vs. (87,21)), yet allows for precise ground-truth benchmarking. We believe our suite, ranging from CartPole to ReposeCube, provides a diverse set of challenges that serve as fundamental building blocks for more complex domains.
>
> ### **2. Comparison with RoboCLIP and FuRL**
>
> We would like to state that direct comparisons to RoboCLIP [4] and FuRL [5] are fundamentally mismatched:
>
> - **RoboCLIP**: This method uses a video-based VLM to provide sparse, trajectory-level, rewards. Our method uses an image-based VLM to provide dense, observation-level, rewards. Because RoboCLIP utilizes temporal video information, it can infer motion goals, whereas our frame-based approach cannot. Additionally, since we focus on the relative performance improvement of SAFT over the underlying model, we cannot apply this analysis to RoboCLIP because CAL and RLR are incompatible with video inputs.
>
> - **FuRL**: This method requires privileged access to ground-truth sparse rewards to fine-tune the VLM. Our method is unsupervised regarding ground-truth rewards. Comparing against FuRL would be unfair as FuRL would be performing the equivalent of "training on the evaluation set". Therefore, VLM-RM (with and without goal-baseline regularization) remains the only methodologically consistent baseline.
>
> ### **3. Clarification on Invariance (Physical State vs. Observation)**
>
> The statement regarding invariance describes the axiomatic definition of the RL problem, not an empirical claim. A ground-truth reward is a function of the underlying physical state. While viewpoint changes or background noise alter the observation, they do not alter the physical state, Thus, the reward must remain identical by definition. Our method simply enforces this axiomatic consistency within the VLM's latent space.
>
> ### **4. Window Size Ablations**
>
> Thank you for pointing this out! We have added window size ablations in Appendix B.4.
>
> ### **5. Pretraining Details**
>
> We have added a dedicated section discussing pretraining in Appendix~A.3. We hope this fully addresses your questions. To summarize, while the main text utilizes regression (MSE) to systematically interpolate between weak and strong underlying models, Appendix B.1 and Appendix B.6 demonstrate that our method also works effectively with off-the-shelf models and PbRL (CE Loss) pretraining.
>
> ### **6. EPIC Distance Introduction within Main Results**
>
> Thank you for pointing this out! We have now reordered the two sections to introduce the EPIC distance first.
>
> ### **7. Proof of Reward Alignment**
>
> We would like to direct your attention to Table 3 rather than Figure 6 for evidence of reward recovery. We observe statistically significant improvements in EPIC distance [1], which **directly** measures the alignment between the learned and ground-truth rewards. Figure 6 merely illustrates the downstream consequence of this alignment: improved policy convergence.
>
> **References**
>
> [1] Gleave et al. "Quantifying Differences in Reward Functions" ICLR 2021
>
> [2] Rocamonde et al. "Vision-language models are zero-shot reward models for reinforcement learning" ICLR 2024
>
> [3] Wang et al. "RL-VLM-F: Reinforcement learning from vision language foundation model feedback" ICML 2024
>
> [4] Sontakke et al. "RoboCLIP: One Demonstration is Enough to Learn Robot Policies" NeurIPS 2023
>
> [5] Fu et al. "FuRL: Visual-language models as fuzzy rewards for reinforcement learning" ICML 2024

---

> > ### Author Response · Authors · 2025-11-21
> > **Official Response to Reviewer dAd2 (Part 2/2)**
> >
> > ### **8. Performance of Goal-Baseline Regularization**
> >
> > We agree with this observation. The poor performance of the goal-baseline regularization [2] remains an open question. The original paper lacks training curves for comparison, and the cited VLM-RM-F paper [3] does not disclose the _α_ value used (or if regularization was used at all). We selected _α=0.5_ based on the original authors' general recommendation ([2], Figure 4a). We hypothesize the performance gap stems from our use of IsaacLab's CartPole, which is a continuous, 3D visual task, making it significantly harder than the discrete, 2D Gym implementation.
> >
> > ### **9. Window Size Mechanism**
> >
> > As detailed in Section 3.1.2, and now ablated in Appendix B.4, the window size _W_ modulates the strength of the RLR loss. Larger window sizes enforce a stronger temporal consistency prior during fine-tuning, and hence directly control its strength.
> >
> > **References**
> >
> > [1] Gleave et al. "Quantifying Differences in Reward Functions" ICLR 2021
> >
> > [2] Rocamonde et al. "Vision-language models are zero-shot reward models for reinforcement learning" ICLR 2024
> >
> > [3] Wang et al. "RL-VLM-F: Reinforcement learning from vision language foundation model feedback" ICML 2024
> >
> > [4] Sontakke et al. "RoboCLIP: One Demonstration is Enough to Learn Robot Policies" NeurIPS 2023
> >
> > [5] Fu et al. "FuRL: Visual-language models as fuzzy rewards for reinforcement learning" ICML 2024

---

### Official Review · Reviewer_WVB3 · 2025-11-01

**Soundness:** 2
**Presentation:** 2
**Contribution:** 2
**Rating:** 4
**Confidence:** 4

**Summary:**

This paper introduces Structure-Aware Fine-Tuning (SAFT), a lightweight method for improving zero-shot Vision-Language Model (VLM) reward models in reinforcement learning. SAFT leverages LoRA-based fine-tuning with augmentation and auxiliary losses. The method is evaluated on classic control and robotic manipulation tasks, demonstrating improved sample efficiency, reduced reliance on human preference labels, and better alignment with ground-truth rewards compared to baseline approaches.

**Strengths:**

1. Combines contrastive learning and Lipschitz regularization in a novel way for fine-tuning VLM reward models;
2. Reduces the need for costly human feedback and enables smaller models to perform comparably to larger ones.

**Weaknesses:**

1. The method assumes the base VLM has a reasonable initial understanding of the task; it may fail if the VLM is fundamentally misaligned.
2. Lack of comparisons on more complex tasks, such as some of the evaluation environments used in [1].

**Questions:**

The authors claim that their proposed method is more robust to visual variations compared to previous approaches [1][2]. Can results on more complex tasks be provided to support this claim? Additionally, would the prior assumptions underlying Lipschitz Regularization remain valid in such tasks? For example, high-dimensional human-robot tasks from or visual RL tasks with background perturbation in [1].

[1] Rocamonde, et al. VISION-LANGUAGE MODELS ARE ZERO-SHOT REWARD MODELS FOR REINFORCEMENT LEARNING. ICLR, 2024.
[2] Fu, et al. FuRL: Visual-Language Models as Fuzzy Rewards for Reinforcement Learning. ICML, 2024.

---

> ### Author Response · Authors · 2025-11-21
> **Official Response to Reviewer WVB3**
>
> We thank the reviewer for their feedback and the time taken to evaluate our work! Below are our responses to the concerns, in the order they were raised.
>
> ### **1. Dependence on Initial VLM Quality**
>
> We agree with your assessment regarding the quality of the base VLM. SAFT is designed to mitigate the brittleness of zero-shot VLM signals, not to recover a fundamentally misaligned world model. Similar to the standard machine learning principle of "Garbage In, Garbage Out", if the underlying model is catastrophically incapable, performance cannot be improved without ground-truth rewards. With this said, we would like to highlight that SAFT offers significant benefits across a wide range of VLM capabilities and never hurts performance, as evidenced in Section 4 as well as Appendix B.1 and B.6.
>
> ### **2. Evaluation Environments (VLM-RM & Humanoid Task)**
>
> Regarding the environments in [1] (VLM-RM), we do indeed evaluate on 2 of their 3 tasks. We purposefully omit their Humanoid task. In [1], this task is adapted to a static goal (involving for example "falling" into a kneeling position) rather than a dynamic motion (like "walking"). This is done because image-based VLMs, which process individual observations and lack a concept of time, are unable to understand temporal dynamics i.e. they can only provide rewards based on a goal state rather than a goal motion. Crucially, this adaptation of the task results in the lack of a well-defined ground-truth reward; the authors simply empirically observe the final policy to judge success. Innate human bias aside that stems from this kind of evaluation, the lack of an unbiased ground-truth reward prevents us from reliably comparing online training performance or EPIC distances.
>
> Instead, we rely on the Allegro Repose Cube task. We highlight that the Allegro Cube is similarly realistic and high-dimensional (Observation dim.: 72, Action dim.: 16) compared to the Humanoid task (Observation dim.: 87, Action dim.: 21). The key difference is that the Cube task possesses an unbiased ground-truth reward, allowing for rigorous evaluation. We show that SAFT outperforms the baseline on this task in Figure 6.
>
> ### **3. Comparison to FuRL**
>
> Regarding the comparison to [2] (FuRL), we would like to state that while VLM-RM is a fair comparison, FuRL is fundamentally different. FuRL utilizes privileged ground-truth knowledge (sparse signals of goal completion) to fine-tune the VLM, effectively using the model as an intermediary to densify sparse rewards. This stands in stark contrast to SAFT, which assumes no ground-truth knowledge or privileged signals in any form. Because FuRL operates under a different supervision setting, comparing with it directly would be an apples-to-oranges comparison. We briefly mention this in Lines 470-472.
>
> ### **4. Lipschitz Regularization & Visual Perturbations**
>
> We acknowledge your concern regarding the RLR prior in high-dimensional tasks and have updated Section 3.1.2 (Lines 245-248) to reflect this limitation. However, regarding visual variations (such as background perturbations), we argue that modeling invariance is crucial to prevent task-irrelevant features from dominating the signal. We believe this specific challenge could effectively be handled by the CAL component of our method.
>
> **References**
>
> [1] Rocamonde et al. "Vision-language models are zero-shot reward models for reinforcement learning" ICLR 2024
>
> [2] Fu et al. "FuRL: Visual-language models as fuzzy rewards for reinforcement learning" ICML 2024

---

### Official Review · Reviewer_kTvJ · 2025-11-01

**Soundness:** 3
**Presentation:** 3
**Contribution:** 2
**Rating:** 6
**Confidence:** 3

**Summary:**

The paper introduces Structure-Aware Fine-Tuning (SAFT), a lightweight LoRA-based method that enhances zero-shot vision-language reward models for reinforcement learning. Instead of relying on human feedback or ground-truth rewards, SAFT enforces structural priors—invariance and proportionality—through two auxiliary objectives: the Contrastive Augmentation Loss (CAL) and Reward Lipschitz Regularization (RLR). These objectives impose smoothness and consistency on the VLM-derived reward landscape. Experiments across classic control and robotic manipulation tasks demonstrate significant gains in sample efficiency, alignment with ground-truth rewards, and reductions in human annotation effort.

**Strengths:**

1. The problem the paper aims to address is highly important. Adjusting VLM-based rewards to better suit downstream tasks without labeled supervision represents a very promising research direction.

2. In terms of performance, the results across the four evaluated tasks effectively support the authors’ claims.

3. The paper is well written and easy to follow.

**Weaknesses:**

I am somewhat concerned that the proposed structure-aware component may be task-specific, tailored to each environment. For more complex or diverse tasks, such manually designed structural rules might not generalize well and could easily become ineffective.

**Questions:**

I also share a concern about whether this algorithm can be successfully applied to more realistic embodied intelligence scenarios, rather than being limited to simulated environments.

---

> ### Author Response · Authors · 2025-11-21
> **Official Response to Reviewer kTvJ**
>
> We thank the reviewer for their time and comments! Below are our responses to the concerns, in the order they were raised.
>
> ### **1. Task-Specificity**
>
> You are correct regarding task-specificity, and we acknowledge this limitation in Section 6. However, we would like to highlight that our manual effort comes at a one-time cost. Once CAL or RLR are configured, training proceeds without intervention. We argue this is a significant advantage over PbRL, which requires continuous, task-specific human feedback throughout training. Additionally, PbRL often sparsifies the reward signal and introduces unintended human bias [1][2].
>
> ### **2. Scalability and Complexity**
>
> Regarding the application to complex embodied intelligence, we acknowledge and do not claim that our method is a silver bullet. That said, we highlight that our method succeeds in the Allegro Cube environment (72-dimensional observation space, 16-dimensional action space), which represents a high-dimensional task by standard benchmarks. Additionally, we argue that the necessity of modeling invariance (via CAL) actually increases with environmental realism/complexity. As tasks become more difficult, task-irrelevant differences threaten to dominate the input signal. Being able to enforce invariances is critical to prevent this. Furthermore, while tasks differ, the atomic nature of augmentations does not. We believe future work can stack a wide variety of positive samples to address significantly more complicated tasks.
>
> **References**
>
> [1] Christiano et al. "Deep reinforcement learning from human preferences" NeurIPS 2017
>
> [2] Tao et al. "Hybrid Reinforcement: When Reward Is Sparse, It's Better to Be Dense" arXiv preprint

---

### Author Response · Authors · 2025-11-21
**General Response to All Reviewers**

## General Response to All Reviewers

We thank the reviewers for their detailed, constructive, and rigorous feedback. We are encouraged that the reviewers found the problem setting interesting and the approach novel. Based on your comments, we have updated the manuscript to improve clarity, reproducibility, and evaluation rigor. Below is a summary of the major changes and a clarification of our core methodological scope.

### **1. Summary of Updates to the Manuscript**

In response to specific reviewer requests, we have added the following to the revised paper:

- **Clarification of Pretraining (Appendix A.3):** A dedicated section detailing the pretraining rationale, the regression target (ground-truth task reward), and the shaping terms used. We clarify that this serves to interpolate between weak and strong base models to measure relative improvement, not to leak ground-truth data.

- **PbRL-Pretrained Evaluation (Appendix B.6):** New results demonstrating SAFT’s effectiveness on a model pretrained via PbRL without access to ground-truth reward values, addressing concerns about dependence on regression-based pretraining.

- **Off-the-Shelf ViT Evaluation (Figure S9):** Training curves for SAFT applied to an off-the-shelf ViT-B16 across all environments over an increased number of seeds.

- **Window Size Ablations (Appendix B.4):** An ablation study on the window size _W_, demonstrating its effect on the strength of the RLR prior.

### **2. Clarification on Scope and "Leakage"**

A primary concern raised (Reviewer _**fyXP**_) was whether pretraining the VLM on ground-truth rewards constitutes "training on the evaluation set". We fundamentally clarify our contribution:

- **Goal:** We do not claim to solve the problem of creating rewards from scratch (as VLM-RMs do [1]). We claim to improve the reward landscape of existing imperfect VLM-RMs.

- **Methodology:** To evaluate this delta, we require base models of varying quality. Pretraining via regression is a controlled mechanism to generate these base models, allowing us to systematically evaluate SAFT across the spectrum of VLM capabilities, from "barely working" to "near-perfect".

- **Validity:** As evidenced by our new experiments on off-the-shelf models and PbRL-pretrained models, SAFT functions independently of how the base model was created.

### **3. Choice of Baselines and Environments**

We reiterate our rationale for the experimental setup:

- **Baselines:** We compare against VLM-RM [1], the only methodologically consistent baseline. We exclude RoboCLIP [2] (video-based/trajectory-level) and FuRL [3] (requires privileged ground-truth sparse rewards) as they operate under fundamentally different assumptions and input modalities.

- **Environments:** We utilize Allegro Repose Cube (72-dim observation, 16-dim action) as our high-dimensional benchmark. We omit the Humanoid task used in prior work because it relies on a modified static goal (e.g. "kneeling") rather than dynamic motion (e.g. "walking") due to VLM limitations, and lacks a rigorous ground-truth reward for quantitative evaluation through policy curves and EPIC distance.

### **4. Reproducibility**

We have provided an anonymous link to the code in our individual response to Reviewer _**fyXP**_ and commit to releasing a fully cleaned and documented version upon acceptance.

We believe these updates and clarifications address the core concerns regarding the validity and fairness of our evaluation. We thank the reviewers once again for all their effort and helping us in strengthening this work!

**References**

[1] Rocamonde et al. "Vision-language models are zero-shot reward models for reinforcement learning" ICLR 2024

[2] Sontakke et al. "RoboCLIP: One Demonstration is Enough to Learn Robot Policies" NeurIPS 2023

[3] Fu et al. "FuRL: Visual-language models as fuzzy rewards for reinforcement learning" ICML 2024

---

### Author Response · Authors · 2025-12-01
**Direct Address to Area Chair**

Dear Area Chair,

We respectfully highlight that Reviewer _**fyXP**_'s score of 0 stems from a factual misunderstanding regarding "data leakage" in our pretraining setup. As detailed in our response, pretraining is a methodological necessity to ensure fair evaluation, used solely to modulate base model quality for evaluating relative improvement, not to construct an unfair comparison.

Crucially, we have added Appendix A.3, Figure S9, and Figure S14 to substantiate this and prevent any further confusion.

---

### Meta-Review · Area_Chair_se3m · 2026-01-07

**Summary:**

The paper proposes Structure-Aware Fine-Tuning (SAFT), a method designed to refine zero-shot reward models derived from VLMs by optimizing LoRA adapters via unsupervised structural objectives. The submission faced a polarized reception, especially a fundamental controversy regarding its experimental methodology and framing. A significant point of contention was the discrepancy between the "Zero-Shot" claim in the title and the primary experimental setup, which utilized base models pre-trained on ground-truth rewards. This led one reviewer to strongly reject the work on the grounds of data leakage and misleading presentation. While the authors successfully demonstrated during the rebuttal that the method functions on off-the-shelf models, this proof-of-concept did not resolve the broader issue that the manuscript's main narrative relies heavily on a confusing and potentially misleading experimental design. Furthermore, concerns regarding the generality of the method persist, as the reliance on manually defined, task-specific augmentations contradicts the promise of an automated, zero-shot pipeline. Consequently, due to these unresolved issues with framing and generality, the paper is not ready for acceptance.

**Reviewer Concerns:**

- Data Leakage (Reviewer fyXP): The reviewer argued that pre-training the VLM on ground-truth rewards invalidates the results and the "Zero-Shot" claim.

- Task Specificity (Reviewers kTvJ, WVB3): Reviewers noted that the structural priors require manual configuration per task, limiting the method's generality.

- Baselines (Reviewers dAd2, WVB3): Reviewers requested comparisons to other VLM reward methods like RoboCLIP or FuRL.

The authors partially addressed the technical aspect of the "leakage" concern during the rebuttal. They provided new results showing that SAFT improves performance without ground-truth access to function. They also clarified that baselines like FuRL use privileged information, making direct comparison unfair. However, he framing and presentation concern remains outstanding. Despite the rebuttal evidence, the main body of the paper still relies on experiments where the base model has seen ground-truth rewards. This creates a confusing and misleading narrative for the reader when contrasted with the title. Furthermore, the reliance on manually defined task-specific priors remains a limitation on the "automated" appeal of the method.

**Reviewer Scores:**

There is no noticeable sign showing any reviewer would raise their ratings based on the visible rebuttal history.

---

### Decision · Program_Chairs · 2026-01-26

Reject